# Machine Learning Approach for Pitch Type Classification Based on Pelvis and Trunk Kinematics Captured with Wearable Sensors

**DOI:** 10.3390/s23239373

**Published:** 2023-11-23

**Authors:** Larisa Gomaz, Celine Bouwmeester, Erik van der Graaff, Bart van Trigt, DirkJan Veeger

**Affiliations:** 1Delft Institute of Applied Mathematics, Delft University of Technology, 2628 CD Delft, The Netherlands; 2BioMechanical Engineering, Faculty of Mechanical, Maritime and Materials Engineering, Delft University of Technology, 2628 CD Delft, The Netherlandsb.vantrigt@tudelft.nl (B.v.T.); h.e.j.veeger@tudelft.nl (D.V.); 3PITCHPERFECT, 4814 GA Breda, The Netherlands; erik.vandergraaff@cir.nl

**Keywords:** baseball, pitching, wearables, classification, pitch types

## Abstract

The large stream of data from wearable devices integrated with sports routines has changed the traditional approach to athletes’ training and performance monitoring. However, one of the challenges of data-driven training is to provide actionable insights tailored to individual training optimization. In baseball, the pitching mechanics and pitch type play an essential role in pitchers’ performance and injury risk management. The optimal manipulation of kinematic and temporal parameters within the kinetic chain can improve the pitcher’s chances of success and discourage the batter’s anticipation of a particular pitch type. Therefore, the aim of this study was to provide a machine learning approach to pitch type classification based on pelvis and trunk peak angular velocity and their separation time recorded using wearable sensors (PITCHPERFECT). The Naive Bayes algorithm showed the best performance in the binary classification task and so did Random Forest in the multiclass classification task. The accuracy of Fastball classification was 71%, whilst the accuracy of the classification of three different pitch types was 61.3%. The outcomes of this study demonstrated the potential for the utilization of wearables in baseball pitching. The automatic detection of pitch types based on pelvis and trunk kinematics may provide actionable insight into pitching performance during training for pitchers of various levels of play.

## 1. Introduction

Data-driven decision-making is establishing itself in training and high-level sports performance. Data made available through game statistics and technology integrated with training routines serve as the input for big data analytics in sports. Data analysis started in many sports disciplines with some form of video analysis. Currently, a variety of different metrics can be extracted and analyzed not only from videos, but also sensors integrated into sleeves, straps, watches, rings, and smart fabrics. For instance, in baseball, for over 100 years, the difference between a slider and a curveball was defined based on previous experience. Following the technological advancements in pitch tracking, the concept of pitch types is quantified and explained by the speed, spin rate, and spin axis of the ball. Information on the ball (Rapsodo), the bat (Blast), and body movement (PITCHPERFECT) has become widely accessible, creating a new flow of data, which are valuable for performance assessment and pitchers’ overall success.

The advancements in wearable technology are changing the traditional approach to athlete training and performance monitoring. Wearables enable measurements in a wide range of settings during training and matches. This removes any practical limitation compared to a lab and offers unlimited athlete availability, which results in high numbers of recorded repetitions. While biomechanical measurements in the lab as well as coaching sessions during training are often limited to one athlete at a time, the utilization of wearables ensures that every pitch thrown by the pitchers is recorded, even the ones during warm-up sessions. The use and collection of data from wearables can be performed by any motivated team that might lack the resources available to professional sports teams, and this enables coaches to retrospectively provide feedback to every pitcher. Such performance tracking in terms of pitch counts enables players to pitch without fatigue, directly adhering to the pitch count limit regulated by the federations in order to limit the workload and prevent shoulder and elbow injuries [1].

Next to the pitch count, the pitching mechanics and pitch type are considered the main factors in pitching training, which are relevant not only for pitchers’ performance, but also for the prevention of injuries [2,3,4,5]. As the pitcher’s response to a given training stimulus is highly individualized [6], continuous and prospective individual monitoring is crucial in managing the effect of the intense training and competition schedule on the pitcher’s performance and health. The use of wearable sensors may provide the opportunity to achieve this.

Information extracted from wearables creates the opportunity to understand the body mechanics of each pitcher on an individual level. Detailed pitch-to-pitch information can help the pitcher learn safe and efficient pitch mechanics. In general, pitching mechanics follow the kinetic chain principle in which the pelvis and trunk serve as a link in the transfer of the momentum generated by the lower extremities to the upper extremities. Efficient proximal-to-distal timing between the pelvis and trunk allows momentum transfer to the ball, resulting in increased throwing velocity [7,8,9]. On the contrary, poor pitching mechanics in combination with the repetitive mechanical strain of throwing through a high pitch count can negatively affect pitching performance and, at the same time, put the pitcher at risk of shoulder and elbow injuries [1,3,4,5].

To translate training success into game success, pitchers need to translate their movement skills into a variation of pitch trajectories. A successful pitcher alters the velocity and trajectory of the ball to keep the batters off balance and discourage their anticipation of a particular pitch type. To obtain a variation of ball trajectory, in theory, the pitcher manipulates the grip on the ball at the release point, which results in different rotations of the ball out of the hand of the pitcher. The particular seams of a baseball lead to air pressure variations around the ball, which creates the bending, curving, or sliding motions of the pitch. It should be noted though that multiple studies have reported differences in the pelvis and trunk kinematics between pitch types [3,10,11,12,13]. From a strategic point of view, a pitcher may want to achieve similar kinematics among all pitch types to make pitch identification difficult for the batters [11]. If that were the case, it would be unlikely that the pitch type could be distinguished from the body mechanics alone. However, the aforementioned studies acquired their data in a lab setting with highly trained individuals. It can be expected that, at lower levels of play, the movement variation within the individual is even higher.

Except the skill difference, there are obvious differences in financial resources and staff availability as well. Although it is common in youth baseball that a volunteer manually counts the amount of pitches, the tracking of the pitch types is very limited, and in particular, off-speed pitches lead to wildly inaccurate manual classifications given the skill level of the person performing the tagging. Therefore, the automatic detection of pitch types might be extremely beneficial, especially for baseball players who cannot afford expensive camera systems and rely on the manual tracking of pitch types. In this context, it should also be noted that off-speed pitches are associated with an increased risk of shoulder and elbow injuries in youth baseball pitchers. In combination with the increased number of pitches per game and the full baseball calendars, pitchers are at risk of not only acute problems, but also overuse injuries in the later stages of their careers [1].

Translating collected wearables data into actionable insights may bridge the gap between scientific knowledge from biomechanical studies and daily practice. We provide a machine learning approach to the utilization of wearables data through pitch type classification based on the pelvis and trunk peak angular velocity and their separation time recorded using body-worn motion sensors. Machine learning methods showed promising results in pitch type classification investigated in similar contexts [14,15,16,17,18,19,20]. Opposed to predicting the next pitch thrown based on the information available prior to that pitch [14,15,16], our approach relies on inclusion of post-delivery features to detect which pitch was thrown purely based on pitching mechanics. Having pitch type readily available on every pitch, in combination with kinematic data, might help us provide insight into pitching technique to baseball pitchers of various levels. On top of that, overview of such performance metrics can be presented to the athletes in real time, enabling players to track their progress throughout the whole season and empowering them to shape the training accordingly.

To the best of our knowledge, this is the first study investigating baseball pitch type detection based on pelvis and trunk kinematics during pitching and, moreover, based on such data obtained from wearables. This approach allows for workload monitoring, which is important for maintaining safe and efficient pitching performance during the full course of the season. Therefore, this study aims to establish the methodology for pitch type classification based on biomechanical input from wearables by comparing performance of the various classification algorithms.

## 2. Materials and Methods

### 2.1. Participants

Out of 24 pitchers initially participating in the measurements, 19 pitchers were included in this study (age 18.5 ± 3.7 years, height 178.3 ± 11.1 m, weight 71.9 ± 18.3 kg, experience 7.3 ± 3.7 years). The participants were members of the elite youth academies of the Royal Dutch Baseball and Softball Federation (KNBSB). The included pitchers were pain- and injury-free during the course of the measurements. This research was conducted in accordance with the Declaration of Helsinki, and the Ethics Committee of the Delft University of Technology approved the measurement protocol (approval no. ETC_TUDelft_1394). Informed consent was signed by the participants or the general manager of the respective baseball academy.

### 2.2. Data Collection and Data Pre-Processing

The data were collected during the pitchers’ regular training at the training facilities of the affiliated baseball academy. To maintain pitching-specific routines, warm-up and pitch count were not standardised. After performing their standard warm-up, the pitchers were instructed to throw a selection of pitch types they usually throw during the game, containing a minimum of three different pitch types. The pitchers followed their own training routine in accordance with the training program set by their pitching coach. The bullpen session consisted of a minimum of 20 pitches from mound toward a catcher at the official distance of 18.45 or 16.45 m, depending on the pitcher’s age.

The pitching motion was recorded using the PITCHPERFECT system (PITCHPERFECT, Breda, The Netherlands) consisting of two synchronised 3-DOF IMUs (Gyroscope ±2000 (°/s)) showed on Figure 1. Sensors were taped with Leukoplast FixoMull® stretch (BSN Medical GmbH, Hamburg, Germany) on processus Xiphoideus on the chest and in the middle of the left and right posterior superior iliac spine on the lower back of the pitcher before starting the bullpen (Figure 2). Pitch types were manually coded by experienced off-field staff members based on the visual inspection, hand signal and pitcher–catcher agreement prior to each throw. The ball velocity (mph) was measured from behind the pitcher with a Pocket radar Ball coach, Model PR1000-BC (Pocket Radar Inc., Santa Rosa, CA, USA). The accuracy of the pitch was noted, distinguishing only between a wild pitch or not, wherein a wild pitch was noted if the catcher was unable to catch the ball with reasonable effort.

The outcome of the PITCHPERFECT system consists of the pelvis and trunk peak angular velocity and the separation time between them. Pre-processing of the raw sensor signal and computing Euclidean norms from the raw data were conducted by the algorithm developed by the manufacturer (PITCHPERFECT, The Netherlands). Details of the algorithm are property of the manufacturer.

In this study, we used a database created by PITCHPERFECT that characterizes each pitch with three features used directly from the system (Table 1). Data were pre-processed and analyzed using the R programming language (version 4.3.1). Data of five players were excluded from the analyses because their peak angular velocity was below the threshold of 400 (°/s) of the PITCHPERFECT system. Individual pitches were included based on three inclusion criteria: (1) the pitch type is a Fastball (FB), Curveball (CU) or Change-up (CH), as they were the most occurring pitch types among the included pitchers; (2) the thrown ball was not a wild pitch; and (3) all three kinematic parameters (*Pelvis*, *Trunk*, *Separation*) were recorded (i.e., sensor clipping did not occur). All continuous features were scaled and centered.

### 2.3. Data Analysis

The automatic detection of pitch types from sensor data is a classification problem. The goal is to learn a mapping from inputs *x* to outputs *y*, where y∈{1,...,C}, with *C* being the number of classes. Inputs *x* are the features (Table 1) and outputs *y* are pitch types, where *C* denotes number of different pitch types.

This study utilized classifiers integrated in the caret package [22] including K-Nearest Neighbors (KNN), Naive Bayes (NB), Random Forest (RF) and Support Vector Machine (SVM). We investigated the performance of the classifiers in both binary and multiclass classification, including additional Logistic Regression (LOGREG) for binary and Multinomial Logistic Regression (MNOM) for the multiclass classification task.

Binary classification is a classification task that has two class labels. In this study, it is used to detect whether the pitch was Fastball or not by classifying recorded pitches in one of the two classes—FB and Other (Figure 3 (left)). Among the recorded pitches, 48.7% were originally labelled as FB and 51.3% as Other.

Multiclass classification refers to classification tasks that have more than two class labels. Unlike binary classification, it classifies non-fastball pitches in different classes and therefore detects whether the pitch was Fastball (FB), Curveball (CU) or Change-up (CH) (Figure 3 (right)). Among the recorded pitches, 48.7% were originally labelled as FB, 26.4% as CH and 24.9% as CU. Due to variations in the number and type of off-speed pitches (CU and CH) among pitchers, the collected data show unequal distribution between classes. Such disparity in the frequencies of the observed classes can have a negative impact on model fitting. A possible solution is to subsample the training data in such a way that mitigates the issue (e.g., under- and oversampling). Hence, to address this issue, the minority classes (CU and CH) were up-sampled so that each class was of equal size.

We set up our training and testing cases following the 80% (training) and 20% (testing) split. To achieve a fair understanding of the generalizability of the classifiers, in the designated training set, Leave-One-Group-Out Cross-Validation (LOGO-CV) was carried out. LOGO-CV is a specific type of *k*-fold cross-validation that utilizes data from each individual pitcher as a test set. The number of folds therefore equals the number of pitchers. For every fold, the model is trained on data from J−1 pitchers and tested on the data from the one left-out pitcher.

The performance of the classifiers is evaluated by four evaluation criteria—Accuracy (Equation 1), Sensitivity (Equation 2), Precision (Equation 3) and F1-score (Equation 4)—which can be calculated from the confusion matrix. The confusion matrix provides a summary of the prediction results of a classification algorithm. In the matrix, the numbers of correct and incorrect predictions are summarised with count values and broken down by each class. The output True Positive (TP) represents the number of positives classified correctly, whereas True Negative (TN) represents the number of correctly classified negatives. False Positive (FP) shows the number of negatives that are classified as positives, whereas False Negative (FN) indicates the number of positives classified as negatives.
(1)$$Accuracy=\frac{TP+TN}{Total\;sample}\;,$$
(2)$$Sensitivity=\frac{TP}{TP+FN}\;,$$
(3)$$Precision=\frac{TP}{TP+FP}\;,$$
(4)$$F1=\frac{2TP}{2TP+FP+FN}\;.$$

The hyper-parameters were tuned using grid search, a default method for optimizing tuning parameters in the caret package [22]. Feature selection was performed using correlation analysis. Since the correlation between the features was low, the models were trained and tested using all variables derived from the PITCHPERFECT system (Table 1).

## 3. Results

A total of 353 pitches thrown by 19 pitchers met the inclusion criteria and were included in the study. Descriptive statistics for binary and multiclass classification is presented in Table 2 and Table 3, respectively. A total of 284 pitches were used for training the models and 69 pitches were used for their testing.

### 3.1. Binary Classification

The performance of the K-Nearest Neighbors, Naive Bayes, Random Forest, Support Vector Machine and Logistic Regression algorithms in the binary classification problem was evaluated using four performance metrics (Equation 1)–(Equation 4). Among the trained classifiers, the Naive Bayes algorithm performed the best in classifying fastballs among the recorded pitches. The confusion matrix seen in Figure 4 shows the summary of the prediction performance for Naive Bayes (Accuracy = 71.0%, Precision = 71.9%, Sensitivity = 67.6%, F1-score = 69.7%). The accuracy of the NB algorithm was 7.2% higher than for KNN, 1.4% higher than for RF, 5.8% higher than for SVM and 20.3% higher than for LOGREG. The sensitivity of the RF algorithm is 11.8% higher than for KNN, 3% higher than for NB, 5.9% higher than for SVM and 17.7% higher than for LOGREG. The precision of the NB algorithm was 7.4% higher than for KNN, 3.3% higher than for RF, 7.2% higher than for SVM and 21.9% higher than for LOGREG. The F1-score of the NB algorithm was 8.2% higher than for KNN, 0.1% higher than for RF, 5.0% higher than for SVM and 18.3% higher than for LOGREG. The confusion matrices with corresponding performance metrics of the remaining algorithms are shown in Appendix A.

### 3.2. Multiclass Classification

The four metrics are used to evaluate the performance of the K-Nearest Neighbors, Naive Bayes, Random Forest, Support Vector Machine and Multinomial logistic regression algorithms in the multiclass classification problem. Among the trained classifiers, the Random Forest algorithm performed the best in classifying pitches in three different classes of pitch types (FB, CH and CU). The confusion matrix seen in Figure 5 shows the summary of prediction performance for Random Forest. The accuracy of the RF algorithm was at 52.2%, which is 7.2% higher than for KNN, 7.2% higher than for NB, 11.6% higher than for SVM and 8.7% higher than for MNOM. The confusion matrices with corresponding performance metrics of the remaining algorithms are shown in Appendix B. Performance metrics of the Random Forest algorithm are reported in Table 4.

## 4. Discussion

The aim of this study was to establish a methodology for pitch type classification based on biomechanical input from wearables. We used pelvis and trunk peak angular velocity and separation time between them as an input and evaluated the performance of five machine learning classifiers in the binary and multiclass classification task. The Naive Bayes algorithm showed the best performance in classifying Fastballs with an accuracy of 71%. Furthermore, in the classification of pitch types as Fastball, Curveball or Change-up, the Random Forest algorithm performed the best with an average accuracy of 61.3% over those three pitch types.

Binary classification was used to detect whether the pitch was Fastball or not. Fastball can be considered a "normal" throw. Fastball is the most common pitch type thrown, specifically among youth pitchers. This has to do with the physical development of youth pitchers where the Fastball pitch is used to learn proper body mechanics and throwing accuracy before learning more demanding off-speed pitches. Therefore, to explore the possibility of pitch type classification based on pitching mechanics, it makes sense to first investigate whether we can detect fastballs. Previous studies that used a binary approach for pitch type prediction focused on predicting whether the next pitch will be Fastball rather than detecting whether Fastball was thrown [15,19]. They used pre-pitch ball data as an input, which resulted in accuracies of 70% [15] and 77.45% [19]. Even though such approach offers benefits for choosing the right strategy, it does not contribute to the pitch tracking as part of the workload monitoring for an individual pitcher.

The multiclass classification task classified recorded pitch types into three categories—Fastball (FB), Change-up (CH) and Curveball (CU). It serves as a base for pitch tracking and detects different pitch types thrown. The Random Forest algorithm performed the best with a 50.0% accuracy in classifying CH, a 60.0% accuracy in classifying CU and a 73.9% accuracy in classifying FB. The performance metrics reported in Table 4 show the performance of the RF classifier for each pitch type versus the rest. Multiclass classification has been a subject of several studies before, focusing on pitch type classification based on pre-pitch ball data. Compared to the accuracy of the Random Forest algorithm revealed in this paper, those studies reported higher predictive accuracies, from 74.5% [20] for the SVM algorithm to 93.63% for the KNN algorithm with Manhattan distance [17,18]. This may be due to the sensitive nature of wearable data and inconsistent pitching mechanics of different pitchers among various pitch types. The feature importance for the Random Forest multiclass classifier revealed that the pitcher’s pelvis peak angular velocity is considered most important for the pitch type classification task, whereas the trunk peak angular velocity is considered the least important (Figure A9).

Although we are confident that the proposed methodology could be key to predict pitch type based on biomechanical data from wearables, the reported accuracies leave much to desire. One limitation of this study was that the amount of collected data was low (*n* = 353). The proof of methodology provided in this paper could serve for a study on a larger scale. Additionally, due to the small sample size of individual pitchers, we were not able to perform the classification of pitch types per individuals. The data from the pitchers have a hierarchical structure, suggesting that pitching mechanics [21] as well as pitch kinematics [23] among different throws are more similar for an individual pitcher compared to others. Therefore, it may be sensible to classify pitch types for individual pitchers. Pitch type prediction by pitch count and by pitcher showed improved performance in the prediction of the next pitch the pitcher will throw based on features available from the previous throws [15,16,18]. Our study would have benefited from longitudinal data collection including kinematic data during the full season. This would allow us to perform classification tasks for different pitch types for individual pitchers. Moreover, matching pitching kinematics with ball speed data may also increase the accuracy of the model.

To the best of our knowledge, this is the first study that uses biomechanical data from wearables to predict pitch types, and thus enriches the available data from an easy-to-use motion sensor system. It is important to clarify that this method is proposed for the classification of the pitch thrown and not the prediction of the next pitch. Pitch prediction uses information available prior the pitch to judge which pitch can be expected. However, pitch type classification uses information available post pitch to determine which pitch type was thrown. Previous studies used post-pitch data from PITCHf/x describing the characteristics of the ball from when it leaves the hand of the pitcher until it crosses the home plate [17,18]. Defining the pitch type from ball flight data is related to the inherent need of redefining pitch types. Traditional pitch type description is not sufficient any longer, with the newly available data in professional pitching. Our methodology aims to expand this knowledge to situations such as youth baseball, where expensive PITCHf/x systems are not prevalent.

The proposed classification method, based on a limited amount of data from youth baseball pitches, shows promising performance in predicting Fastball vs. off-speed pitches. Application of this binary classification method in youth baseball training can create a major advantage for the development of individual players. Since nowadays pitch count is the only variable that is noted, and mostly manually recorded, the automatic tracking of pitch counts, biomechanical data and pitch types can be of great value to coaches and players. Given that youth players are still learning how to throw different pitch types and their susceptibility to injuries is higher when throwing off-speed pitches [1], implementing the proposed methods in baseball practice may provide a wealth of information relevant for both pitchers and coaches in those situations.

Implementing similar technologies for elite athletes’ training could benefit from the aforementioned suggestions to improve the accuracy of the multiclass classification model. However, further studies should determine the necessity of such a system since high-level players often have access to other resources that can measure or calculate pitch trajectory. Indirect pitch type prediction may thus not be needed for players at a high level with many resources at their disposal.

## 5. Conclusions

The accessibility of wearable sensors for performance tracking during both training and games represents a new source of large amounts of data that need powerful algorithms for their analysis, resulting in actionable insights relevant for pitchers’ performance and injury risk management. This study established machine learning methods for the detection of the pitch type that was thrown based on pitching mechanics recorded with wearables. The Naive Bayes algorithm showed the best performance in the detection of fastballs, whereas the Random Forest algorithm performed best in the multiclass (FB vs. CH vs. CU) classification task. While these findings demonstrate the potential for the utilisation of wearables in baseball pitching, further development of the classification algorithm, as well as longitudinal data collection, is required. Providing insight into pitch count, pitching mechanics and pitch type enables pitchers to throw safely and efficiently. Through automatic tracking of pitch types, every pitch is counted. Thus, monitoring pitching mechanics and providing an informative feedback to the pitchers may lead to safe and efficient pitching and increase a pitcher’s chances of success.

## Figures and Tables

**Figure 1 sensors-23-09373-f001:**
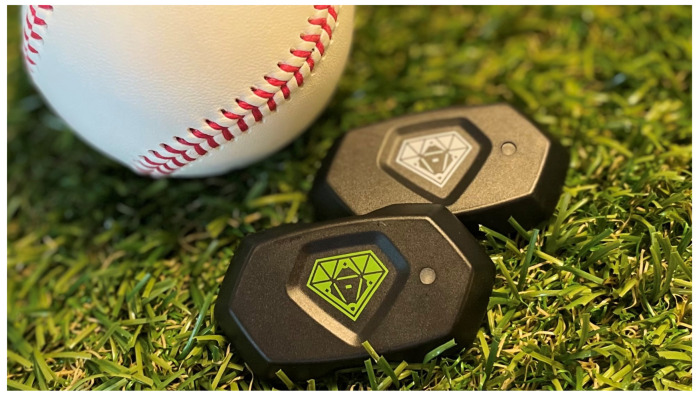
Pitch Perfect sensor system for measuring pelvis and trunk kinematics and separation time between them.

**Figure 2 sensors-23-09373-f002:**
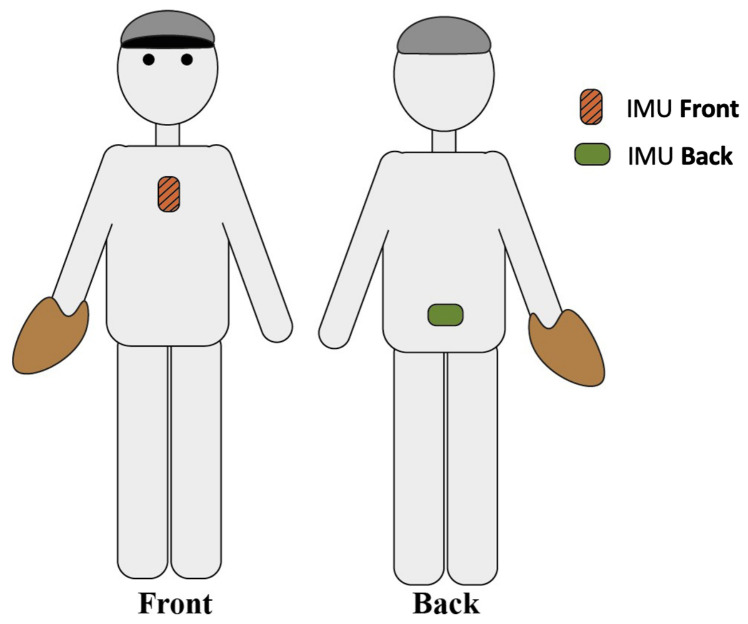
Placement of the sensors. Figure adopted from the study of Gomaz et al. [21].

**Figure 3 sensors-23-09373-f003:**
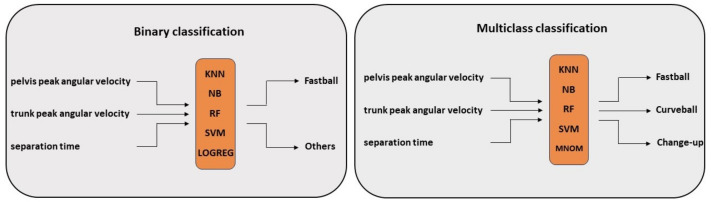
The baseball pitch type classification approaches. (**Left**) The binary classification approach classifies pitch types into two categories—Fastball and Others—based on input from wearables (pelvis and trunk peak angular velocities and separation time). (**Right**) The multiclass classification approach classifies pitch types into three categories—Fastball, Curveball and Change-up—based on input from wearables (pelvis and trunk peak angular velocities and separation time). Both approaches used four classifiers—K-Nearest Neighbors (KNN), Naive Bayes (NB), Random Forest (RF) and Support Vector Machine (SVM)—to assess their classification performance, including additional logistic regression (LOGREG) for binary and multinomial logistic regression (MNOM) for the multiclass classification task.

**Figure 4 sensors-23-09373-f004:**
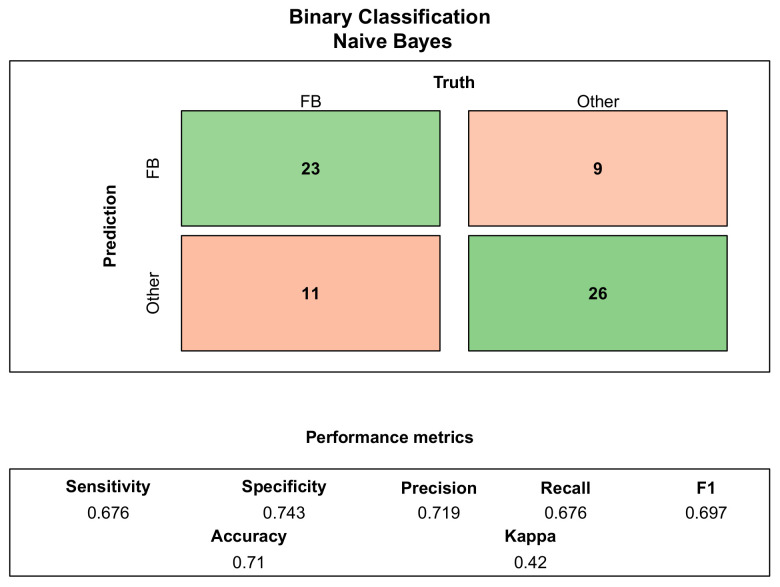
Two-class confusion matrix summarizing the performance of Naive Bayes in classification of fastballs.

**Figure 5 sensors-23-09373-f005:**
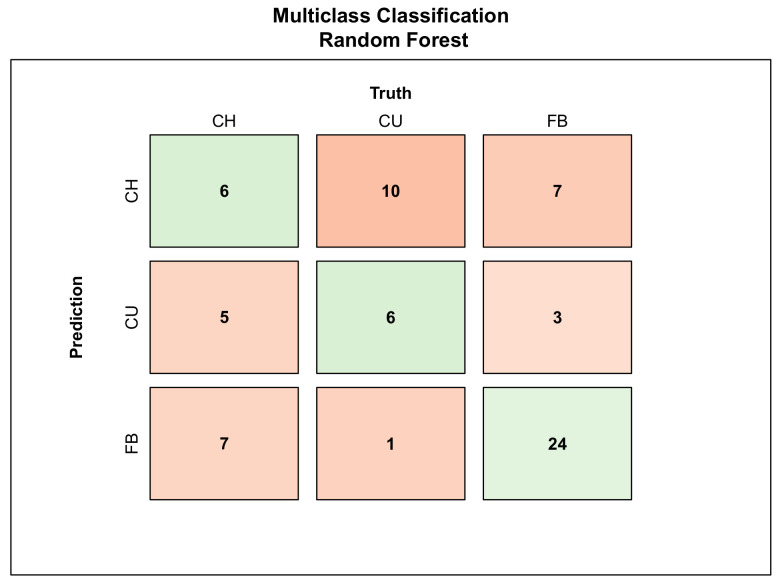
Three-class confusion matrix summarizing the performance of Random Forest by class in classification of baseball pitch types.

**Table 1 sensors-23-09373-t001:** Included features for pitch type classification.

Features	Definitions
Pelvis (°/s)	Pelvis peak angular velocity available directly from PITCHPERFECT.
Trunk (°/s)	Trunk peak angular velocity available directly from PITCHPERFECT.
Separation (ms)	The timing between pelvis and trunk peak angular velocity, available directly from PITCHPERFECT.

**Table 2 sensors-23-09373-t002:** Descriptive statistics for binary classification.

Features	FB		Other	
	(*n* = 172)		(*n* = 181)	
	Mean	SD	Mean	SD
Pelvis (°/s)	737	138	695	120
Trunk (°/s)	799	228	827	262
Separation (s)	0.03	0.13	0.06	0.13
Speed (m/s)	33.1	3.82	28.6	3.81

**Table 3 sensors-23-09373-t003:** Descriptive statistics for multiclass classification.

Features	CH		CU		FB	
	(*n* = 93)		(*n* = 88)		(*n* = 172)	
	Mean	SD	Mean	SD	Mean	SD
Pelvis (°/s)	708	129	681	109	737	138
Trunk (°/s)	831	277	823	247	799	228
Separation (s)	0.06	0.15	0.06	0.11	0.03	0.13
Speed (m/s)	29.9	3.72	27.2	3.40	33.1	3.82

**Table 4 sensors-23-09373-t004:** Performance metrics of multiclass Random Forest in classification of three different pitch types.

Class	Accuracy	Sensitivity	Precision	F1
CH	0.500	0.333	0.261	0.293
CU	0.600	0.353	0.429	0.387
FB	0.739	0.706	0.750	0.727

## Data Availability

The data presented in this study are openly available in 4TU. ResearchData repository at https://data.4tu.nl/datasets/f86ba220-08a1-4fa0-89a9-d8995790675b (accessed on 6 November 2023). The code is available at https://data.4tu.nl/datasets/e339176b-0ecd-48e5-bc7e-9b587c0a8959 (accessed on 9 November 2023).

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
