# Peer review of "Machine Learning Approach for Pitch Type Classification Based on Pelvis and Trunk Kinematics Captured with Wearable Sensors"

_sensors, 2023, doi:10.3390/s23239373_

Round 1
Reviewer 1 Report
Comments and Suggestions for Authors
Dear authors,
I have reviewed your article and believe it has the potential for publication. However, there are several issues that need to be addressed before it can be considered.
Major Issues:
1. Introduction:
The practical need for this study isn't entirely clear. How could this approach benefit coaches in improving training strategies?
2. Materials and Methods:
This section is lacking in detail, particularly regarding the construction of your machine learning models.
a. Why were five pitchers excluded from the study, and what were the exclusion criteria?
b. Please provide a more detailed description of the normalization process for continuous features. Did you normalize before training all the machine learning models?
c. Given the relatively small dataset, consider including multinomial logistic regression in addition to complex algorithms like Random Forest. Justify the need for powerful algorithms when simpler regression models might suffice.
d. Describe the hyperparameter tuning process.
e. Explain the rationale behind evaluating both binary and multiclass classification. Clarify why you compared fastball vs. other pitches in binary classification and whether fastballs require more attention.
f. Present a feature importance analysis to identify variables contributing significantly to the model's predictive ability.
3. Results:
You adopted a leave-one-out cross-validation approach but presented results from only one iteration. Results from all iterations should be considered, and performance metrics should be reported as averages with standard deviations.
4. Discussion and Conclusion:
In these sections, clarify the advantages of using machine learning techniques and how they can enhance training strategies and reduce the risk of injuries, as mentioned earlier. Currently, there isn't a clear advantage to relying on machine learning.
Minor Issues:
Line 113: Consolidate references in one section.
Line 117: When presenting pitch types for the first time, specify the meanings of FB, CU, and CH.
Figure 2: Consider omitting this figure, as it may not be necessary for readers.
Thank you for considering these suggestions for improving your article. I believe that with these revisions, your work will have a stronger chance of being accepted for publication. If you have any questions or need further clarification on any of the points raised, please don't hesitate to reach out.
Comments on the Quality of English Language
A review by a native English speaker would help improve the understanding of the text.
Author Response
Dear Reviewer,
Thank you for reading our manuscript and for your valuable comments and feedback. We revised our manuscript following your suggestions. Please see the attachment.

Reviewer 2 Report
Comments and Suggestions for Authors
-
A comparison of the cost of the sensor used and the alternatives available in the market would benefit the work
-
The motivation seems weak; how the classification of the correct pitch improves player training is not clear.
-
Page 1 - 80. Why 5 participants were excluded?
-
The leave one out validation approach is problematic in this case. The split between train and test should be done by separating the individuals (eg. train 18 participants and test on the last one). Otherwise, the models are likely overfitting the specific individual.
-
No discussion on the model parameters and hyperparametrisation process.
-
Why, with 353 measurements, Figure 4 shows only 70 samples in the CM.
-
Not making the data and the code available is a major drawback of the paper.
-
While accuracies over different datasets are not directly comparable, it’ll be interesting to extend the discussion with more comparison between the current approach and the literature
-
Unbalancing of the training data can be addressed with under- or oversampling approaches.
-
Fix the author's contributions; it still contains the guidelines.
-
“The accuracy of the fastball classification (74%) was higher than for 239 the classification of different pitch types (65%). “ This is obvious since the first one is a variation of the second where two classes are aggregated.
Author Response

(The authors gave the same response as above.)

Round 2
Reviewer 1 Report
Comments and Suggestions for Authors
Dear authors,
I commend your diligent efforts in enhancing the article. Overall, it demonstrates significant improvements. However, I would like to draw your attention to a minor concern:
In Section 2-f, I had requested a feature importance analysis to gain insights into the features considered most influential for the prediction task. The request was for an analysis of feature importance, not feature selection.
Furthermore, it might be beneficial to explore the following article for additional insights if you find it relevant to your research:
Mandorino, M., Tessitore, A., Leduc, C., Persichetti, V., Morabito, M., & Lacome, M. (2023). A New Approach to Quantify Soccer Players’ Readiness through Machine Learning Techniques. Applied Sciences, 13(15), 8808.
Comments on the Quality of English Language
Minor editing
Author Response
Dear Reviewer,
Thank you for reading our manuscript and for your valuable feedback. We revised our manuscript following your suggestions.
Please see attached our adjustments and answers to your comments.

Reviewer 2 Report
Comments and Suggestions for Authors
I'm satisfied with the edits done by the authors. I still think the code should be made publicly available to better evaluate the machine learning part.
Author Response
Dear Reviewer,
Thank you for reading our manuscript and for your valuable feedback. We revised our manuscript following your suggestions.
Please see attached adjustments and answers to your comments.
